# Harnessing the Power of Large Language Models for Empathetic Response Generation: Empirical Investigations and Improvements

**Yushan Qian, Wei-Nan Zhang, Ting Liu**[*]
[*]Research Center for Social Computing and Information Retrieval
Harbin Institute of Technology, China
{ysqian, wnzhang, tliu}@ir.hit.edu.cn

## Abstract

Empathetic dialogue is an indispensable part of building harmonious social relationships and contributes to the development of a helpful AI. Previous approaches are mainly based on fine small-scale language models. With the advent of ChatGPT, the application effect of large language models (LLMs) in this field has attracted great attention. This work empirically investigates the performance of LLMs in generating empathetic responses and proposes three improvement methods of semantically similar in-context learning, two-stage interactive generation, and combination with the knowledge base. Extensive experiments show that LLMs can significantly benefit from our proposed methods and is able to achieve state-of-the-art performance in both automatic and human evaluations. Additionally, we explore the possibility of GPT-4 simulating human evaluators.

## 1 Introduction

Empathetic dialogue plays an essential role in building harmonious social relationships (Zech and Rimé, 2005). The task of empathetic response generation involves understanding the user's experiences and feelings, and generating appropriate responses (Keskin, 2014; Rashkin et al., 2019). Using dialogue systems to provide empathetic responses has advantages such as easy access and no time constraints (Sharma et al., 2020). Figure 1 shows an example of the empathetic dialogue from the benchmark dataset.

Most previous researchers have established elaborately designed models based on reliable theoretical knowledge (Lin et al., 2019; Majumder et al., 2020; Li et al., 2020; Sabour et al., 2022; Li et al., 2022; Zhou et al., 2022). However, the basic models used are mostly small in scale. Recently, large language models (LLMs) (Brown et al., 2020; Chowdhery et al., 2022; Touvron et al., 2023) have

_________
[*] Corresponding author.

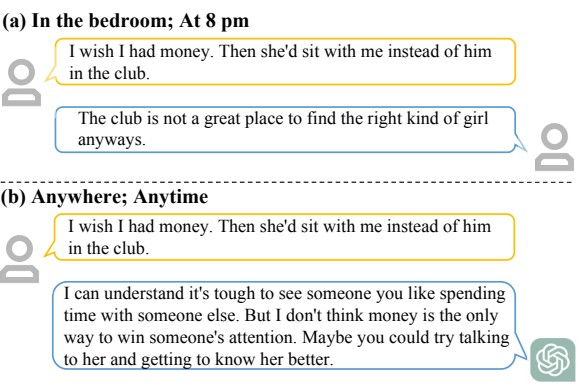

Figure 1: An example of empathetic dialogue from the EMPATHETICDIALOGUES dataset.

been widely used in natural language processing (NLP) with superior performance. In particular, the emergence of ChatGPT has elicited substantial attention and interest in academia and industry, and it has demonstrated extraordinary performance in a variety of tasks, especially dialogue generation. These LLMs are trained on a large amount of corpora, encompassing a wealth of knowledge. In specific tasks, even without fine-tuning, outstanding performance can be achieved by adopting some gradient-free techniques (Brown et al., 2020; Wei et al., 2022) (e.g., in-context learning (ICL)). Therefore, it is necessary to empirically explore the performance of LLMs on specific domains, as the methods of solving problems may undergo significant changes. There have been some initial attempts (Roller et al., 2021; Lee et al., 2022) to apply LLMs to empathetic response generation. However, their approaches mainly focus on pre-training or fine-tuning on the training data, or simply exploring the capability of a single model.

To investigate the capability of LLMs in empathetic response generation, this work empirically studies the performance of LLMs on the empathetic dialogue benchmark dataset. We first compare LLMs in the zero-shot and few-shot ICL set-

tings with a large number of baseline models. Surprisingly, the performance of the GPT-3.5 series of LLMs with in-context learning settings has comprehensively surpassed state-of-the-art models. This reveals that the paradigm shift brought by LLMs also applies to empathetic dialogue. Furthermore, based on the best performance LLM setting, we propose three possible methods to improve its performance. Specifically, improvement via semantically similar in-context learning, two-stage interactive generation, and combination with the knowledge base. Extensive automatic and human evaluation experiments show that LLMs can benefit from our proposed methods, which can generate more empathetic, coherent, and informative responses. In addition, although human evaluation is crucial in empathetic dialogue, its associated costs and time consumption are enormous. In view of the outstanding performance of LLMs on empathetic response generation, we attempt to use GPT-4 (OpenAI, 2023) to simulate human evaluators to evaluate the results. The Spearman and Kendall-Tau correlation results indicate that GPT-4 has the potential to be a substitute for human evaluators.

Our contributions are summarized as follows:

(1) To the best of our knowledge, it is the first comprehensive empirical investigation on the performance of LLMs represented by ChatGPT on empathetic dialogue.

(2) We construct a unified prompt template for the empathetic response generation, and LLMs guided by the template achieve outstanding performance.

(3) We propose three targeted improvement methods, and sufficient experiments demonstrate their effectiveness.

(4) We explore the possibility of GPT-4 simulating human evaluators.

## 2 Related Work

### 2.1 Empathetic Response Generation

Empathy is a complex multi-dimensional structure in psychology and has rich forms in practice (Davis et al., 1980). At present, two main forms of modeling empathy are affective empathy and cognitive empathy (Davis, 1983). Affective empathy oriented methods include mixture of experts (Lin et al., 2019), emotion mimicry (Majumder et al., 2020), and multi-resolution user feedback (Li et al., 2020). Cognitive empathy oriented methods include emotion causes (Gao et al., 2021; Kim et al., 2021;

Qian et al., 2023a), empathetic intents (Welivita and Pu, 2020; Chen et al., 2022), external knowledge (Li et al., 2022; Sabour et al., 2022; Zhou et al., 2022; Cai et al., 2023). Besides, Wang et al. (2022) models the interaction between knowledge and emotion, Zhao et al. (2022) considers self-other awareness, Bi et al. (2023) and Kim et al. (2022) propose multi-grained and fine-grained levels, respectively. However, most of researchers design elaborate small-scale models and the application of LLMs represented by ChatGPT in the empathetic dialogue has not been fully empirically explored.

### 2.2 Large Language Models

Large language models (LLMs) such as GPT-3 (Brown et al., 2020), PaLM (Chowdhery et al., 2022), and LLaMA (Touvron et al., 2023) are pretrained on extensive and large amounts of data, and their tens or hundreds of billions of parameters contain a lot of knowledge. Recently, in combination with new training techniques such as reinforcement learning from human feedback (RLHF) and instruction tuning (Ouyang et al., 2022), the capabilities of LLMs have made a qualitative leap. For example, the emergence of ChatGPT has aroused great interest in academia and industry, demonstrating extraordinary capabilities in a variety of tasks. GPT-4 (OpenAI, 2023) has given some researchers a glimpse of the spark of artificial general intelligence (AGI) (Bubeck et al., 2023). The powerful in-context learning capabilities of LLMs have also led to a paradigm shift.

There are some preliminary attempts to apply LLMs to empathetic dialogue. Blenderbot (Roller et al., 2021) can properly demonstrate empathy through the introduction of the blended skill talk (BST) setup and the correct choice of generation strategies. However, the implementation of empathy is mainly through pre-training with high-quality data and does not utilize the emerging ICL capabilities of LLMs. Lee et al. (2022) explores the performance of GPT-3 to generate empathetic responses with prompt-based in-context learning capabilities. However, they only explore GPT-3, and the capabilities of LLMs have greatly improved with the emergence of new training technologies.

## 3 Methodology

### 3.1 Overview

Formally, the dialogue context is alternate utterances between the speaker and the listener, defined

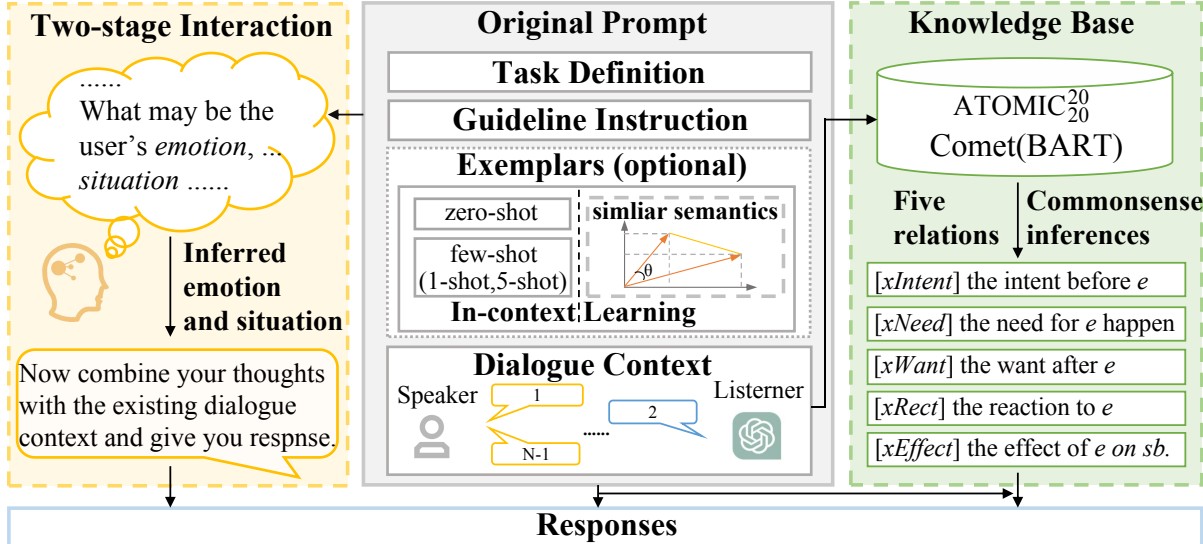

Figure 2: The overall architecture and flow of our proposed methods for LLMs in empathetic dialogue generation.

as $C = \{U_1, U_2, \ldots, U_{n-1}\}$, where $U_i$ represents the $i$-th utterance and $n$ denotes the number of utterances in a dialogue. Our goal is to play the role of the listener and generate the empathetic, coherent and informative response $Y$, which is $U_n$.

The overview of our proposed methods is illustrated in Figure 2, which includes the devised unified template of empathetic response generation and three improvement methods. The left part describes the improvement via two-stage interactive generation, the middle part displays the components of the devised unified template and the improvement via semantically similar in-context learning, and the right part illustrates details of improvement via the knowledge base.

## 3.2 Preliminary Exploration

LLMs possess the ability of in-context learning (ICL) (Brown et al., 2020), by providing task instructions and some examples to LLMs, they can perform related tasks without fine-tuning. This capability significantly alleviates the demand for training data. We first investigate the performance of LLMs on zero-shot ICL and few-shot ICL in empathetic response generation. Since different prompts may affect performance, we strive to maintain a consistent style when designing prompts. The devised prompt template for empathetic dialogue consists of the following components:

Task Definition + Guideline Instruction + Exemplars (optional) + Dialogue Context

Among them, **Task definition** is the researchers' standard definition of the task. **Guideline Instruction** is the instruction we expect the model to follow. **Exemplars** are complete instances of dialogs used to help models better understand the task. **Dialogue Context** is the historical dialogue between the speaker and the listener, and the last sentence is the speaker's utterance. Our goal is to let the dialogue system generate the next round of the listener's utterance. The example of the prompt template is listed in Appendix A.

In the preliminary experimental exploration, we perform three groups of settings.

**0-shot.** This represents a straightforward approach to leverage LLMs for empathetic response generation, which means there are no Exemplars.

**1-shot.** We randomly sample a complete dialogue from the training set as the Exemplar.

**5-shot.** We randomly sample five complete dialogues from the training set as the Exemplars.

## 3.3 Advanced Exploration

In this section, we gradually introduce three methods to improve the performance of LLMs in generating empathetic responses.

### 3.3.1 Improvement via Semantically Similar In-Context Learning

As Liu et al. (2022) argues, a small amount of carefully selected data can greatly improve the performance of LLMs without a large amount of data. We reasonably speculate that in addition to the number of instances, the quality of the instances will

also have an impact on the model's performance. Therefore, when choosing in-context instances, we select a few instances from the training set whose dialogue context semantics are closest to those in the test set.

Specifically, we concatenate the dialogue context of each instance into a long sentence and use a sentence encoder to obtain its vector representation, which represents the semantics of each instance's dialogue context. For the sentence encoder, we adpot the "all-mpnet-base-v2" version of sentence-transformers (Reimers and Gurevych, 2019).[1] It maps sentences to a 768 dimensional dense vector space. The sentence embedding model was trained on very large sentence level datasets using a self-supervised contrastive learning objective. The similarity between semantics is measured by calculating the cosine similarity between the vector representations of two sentences:

$$S = U_1 \oplus U_2 \oplus ... \oplus U_{n-1}, \quad (1)$$

$$E_{train} = \text{Enc}_{sen}(S_{train}), \quad (2)$$

$$E_{test} = \text{Enc}_{sen}(S_{test}), \quad (3)$$

$$\text{Sim}(S_{train}, S_{test}) = \frac{E_{train} \cdot E_{test}}{|E_{train}||E_{test}|}, \quad (4)$$

where $E_{train}$, $E_{test}$ are the sentence encodings of the dialogue context in the training and test set, respectively. $\text{Sim}()$ is used to calculate the similarity of two sentence vectors.

### 3.3.2 Improvement via Two-stage Interactive Generation

In the setting of the empathetic dialogue task, the dialogue system needs to infer what the speaker's emotion is and what the situation is that caused this emotion, so as to provide an appropriate response. Inspired by some pipeline methods in open-domain dialogue (Song et al., 2020; Qian et al., 2023b) and combining the characteristics of empathetic response generation, we can conduct the multi-turn interaction to let LLMs generate appropriate responses. Specifically, in the first stage, we let LLMs speculate on the user's emotional state and experienced situation. In the second stage, the inferred intermediate results are used as input to continue calling LLMs to obtain the final response. Formally, we can express it as:

$$P(Y|T,G,C) = P(e,s|T,G,C)P(Y|e,s), \quad (5)$$

[1]https://huggingface.co/sentence-transformers/all-mpnet-base-v2

where $T$, $G$, $C$ are Task Definition, Guideline Instruction and Dialogue Context, respectively. $e$ and $s$ represent the inferred emotion and situation, respectively.

The prompts we designed in two stages are:

> [The first stage]
> *"Don't rush to reply yet, let's think step by step. Based on the existing dialogue, what may be the user's emotion, and according to his description, what may be the situation when he feels this way?"*
>
> [The second stage]
> *"Now combine your thoughts with the existing dialogue context and give your response."*

The model's thought process during the intermediate step is a basis for generating the final response, enhancing the model's interpretability. At the same time, it also facilitates the analysis of the impact of different key factors (such as emotions and situations) on the final result. Moreover, clearer error analysis is possible when generating responses do not work well.

### 3.3.3 Improvement via Knowledge Base

Merely inferring the speaker's emotions and situation from the historical dialogue is insufficient. A direct evidence is that the response has almost no non-stopword overlapping with the dialogue history in the benchmark dataset (Li et al., 2022). Dialogue systems need more external knowledge to conduct empathetic dialogue. LLMs store a large amount of knowledge through weights, so when performing specific tasks, how to better stimulate the use of relevant knowledge is crucial for improving the effect. An alternative solution is to fine-tune LLMs for specific tasks, but this process usually requires expensive hardware, time, and training data. Inspired by recent work on empathetic dialogue (Sabour et al., 2022), we augment the dialogue context with the commonsense knowledge graph, dynamically utilize external information to stimulate the relevant knowledge encoded by LLMs, and thus generate more empathetic responses.

Similar to Sabour et al. (2022); Zhou et al. (2022), we adopt the commonsense knowledge base ATOMIC$_{20}^{20}$ (Hwang et al., 2021), which contains knowledge not readily available in pre-trained language models, and can generate accurate and representative knowledge for unseen entities and

events. The $\textsc{Atomic}_{20}^{20}$ knowledge base is in the form of event, relation type and inferred knowledge triples. We adopt the BART version of COMET (Hwang et al., 2021) trained on this knowledge base to generate commonsense inferences of five relations (*xIntent*, *xNeed*, *xWant*, *xEffect*, *xReact*) for dialogue contexts. We also design an algorithm to construct the suitable prompt, which can dynamically concatenate the corresponding commonsense inferences according to different dialogue contexts, enriching the input representation, so as to stimulate the relevant knowledge of LLMs more accurately and generate more appropriate responses:

$$CS_r = \text{COMET}_{\text{BART}}(C, r), \qquad (6)$$

$$CS_{kno} = \underset{R}{\oplus} CS_r, \qquad (7)$$

$$C' = C + CS_{kno}, \qquad (8)$$

$$P(Y) = P(Y|T, G, C'), \qquad (9)$$

where $r$ represents the relation type, $r \in R$, and $R = \{xIntent, xNeed, xWant, xEffect, xReact\}$. $CS_{kno}$ is the concatenated external knowledge.

## 4 Experimental Setup

### 4.1 Dataset

EMPATHETICDIALOGUES (Rashkin et al., 2019) is a large-scale benchmark dataset of multi-turn empathetic dialogue in English. Each dialogue in the dataset has an emotion label (32 types in total) and the situation corresponding to the emotion label. The speaker talks about their situation and the listener attempts to understand the speaker's feelings and reply appropriately.

### 4.2 Compared Models

We compare LLMs with the recent state-of-the-art models: (1) MoEL (Lin et al., 2019). (2) MIME (Majumder et al., 2020). (3) EmpDG (Li et al., 2020). (4) EC (Gao et al., 2021). (5) EmpHi (Chen et al., 2022). (6) KEMP (Li et al., 2022). (7) CEM (Sabour et al., 2022). (8) CASE (Zhou et al., 2022). (9) BlenderBot (Roller et al., 2021). The details of compared models are listed in Appendix B.

### 4.3 Evaluation Metrics

We follow previous related studies, conducting both automatic and human evaluations, and choose as many metrics as possible.

**Automatic Evaluation** We adopt Distinct-n (Dist-1/2) (Li et al., 2016), BERTscore ($P_{\text{BERT}}$, $R_{\text{BERT}}$, $F_{\text{BERT}}$) (Zhang et al., 2020), BLEU-n (B-2/4) (Papineni et al., 2002) as main automatic metrics for the performance of the response generation. Distinct-n measures the proportion of distinct n-grams of the response, which is used for diversity evaluation in open-domain dialogue. BERTScore leverages the pre-trained embeddings from BERT and matches words in candidate and reference sentences by cosine similarity. We employ matching precision, recall and F1 score. BLEU-n measures the similarity and relevance between the generated and golden responses. We don't employ Perplexity (PPL) because there are differences in the vocabulary of multiple models. Additionally, some baseline models perform emotion classification as a part of their training process, we also report the emotion prediction accuracy (Acc).

**Human Evaluation** In human evaluation, we randomly sample 100 dialogues from the testing dataset. Considering both the human labor cost and the reliability of the experiment, we select competitive models from the past year (including state-of-the-art) and BlenderBot as representative baselines. Given the dialogue context and these models' generated responses, we recruit three annotators (majority rule) to assign a score from 1 to 5 (1: not at all, 3: OK, 5: very good) to the generated responses based on the aspects of Empathy, Coherence, Informativity, and Fluency. The four aspects are 1) **Empathy** (Emp): whether the response shows an understanding of the user's feelings and experiences, and expresses appropriately; 2) **Coherence** (Coh): whether the response is coherent and relevant to the context; 3) **Informativity** (Inf): whether the response contains more valuable information; 4) **Fluency** (Flu): whether the response is readable. More details about the human evaluation can be found in Appendix C.

Furthermore, we conduct another human A/B test to directly compare different models, taking into account the variation among different individuals. Following Sabour et al. (2022), we conduct the pairwise preference test based on aspects. Given the context, we pair the responses generated by two different methods and ask annotators to choose the better response based on the context and the above four aspects. If the difference is really not significant, a tie is allowed.

| Models | Dist-1 | Dist-2 | $P_{BERT}$ | $R_{BERT}$ | $F_{BERT}$ | B-2 | B-4 | Acc |
|---|---|---|---|---|---|---|---|---|
| *State-of-the-art baselines* | | | | | | | | |
| MoEL | 0.47 | 2.16 | 0.8557 | 0.8598 | 0.8576 | 6.95 | 1.99 | 30.75 |
| MIME | 0.45 | 1.83 | 0.8529 | 0.8605 | 0.8566 | 6.78 | 1.94 | 31.21 |
| EmpDG | 0.47 | 1.98 | 0.8559 | 0.8620 | 0.8588 | 7.17 | 2.03 | 30.64 |
| EC (Hard) | 1.95 | 9.49 | 0.8409 | 0.8592 | 0.8498 | 6.37 | 1.64 | - |
| EC (Soft) | 1.70 | 8.49 | 0.8443 | 0.8593 | 0.8516 | 6.39 | 1.70 | - |
| EmpHi | 0.88 | 4.21 | 0.8359 | 0.8568 | 0.8459 | 5.05 | 1.24 | - |
| KEMP | 0.66 | 3.26 | 0.8533 | 0.8597 | 0.8564 | 5.99 | 1.82 | 36.57 |
| CEM | 0.64 | 2.86 | 0.8577 | 0.8604 | 0.8589 | 5.64 | 1.70 | 37.81 |
| CASE | 0.66 | 3.26 | 0.8571 | 0.8613 | 0.8591 | 7.90 | 2.41 | 38.92 |
| Blenderbot | 1.63 | 8.41 | 0.8285 | 0.8675 | 0.8474 | 7.10 | 2.27 | - |
| *Large language models* | | | | | | | | |
| GPT-3 (+ 0-shot) | 2.07 | 9.08 | 0.8562 | 0.8610 | 0.8584 | 6.88 | 2.22 | - |
| GPT-3 (+ 1-shot) | 2.45 | 11.24 | 0.8571 | 0.8624 | 0.8596 | 6.71 | 2.16 | - |
| GPT-3 (+ 5-shot) | 2.49 | 11.69 | 0.8611 | 0.8640 | 0.8624 | 8.33 | 2.83 | - |
| GPT-3.5 (+ 0-shot) | 2.37 | 11.84 | 0.8737 | 0.8727 | 0.8731 | 8.51 | 2.80 | - |
| GPT-3.5 (+ 1-shot) | 2.68 | 12.29 | **0.8803** | 0.8678 | 0.8739 | 5.62 | 1.99 | - |
| GPT-3.5 (+ 5-shot) | 2.90 | 14.13 | 0.8801 | 0.8749 | 0.8773 | **9.37** | **3.26** | |
| ChatGPT (+ 0-shot) | 2.72 | 17.12 | 0.8679 | 0.8791 | 0.8733 | 6.19 | 1.86 | - |
| ChatGPT (+ 1-shot) | 2.82 | 17.44 | 0.8703 | 0.8791 | 0.8746 | 6.79 | 2.12 | - |
| ChatGPT (+ 5-shot) | **2.96** | **18.29** | 0.8736 | **0.8816** | **0.8774** | 7.85 | 2.65 | - |

Table 1: Results of automatic evaluation between LLMs and baselines.

## 4.4 Implementation Details

We use OpenAI's GPT family [2] as our LLMs. More specifically, we use the model *gpt-3.5-turbo* provided in the OpenAI API, which is the base model of ChatGPT. We also test with GPT-3 *davinci* and another version of GPT-3.5 (*text-davinci-003*). we set temperature to 0 to make the outputs mostly deterministic in the experiment. We divide the dataset into training, validation, and testing set according to the original paper (Rashkin et al., 2019) with 8:1:1. For a fair comparison, the parameter settings of all SOTA models are consistent with those recommended in their initial paper or code.

## 5 Results and Analysis

### 5.1 Preliminary Exploration Results

Table 1 shows the automatic evaluation results between LLMs and baselines. LLMs significantly outperform existing SOTA baselines and achieve a significant improvement on all automatic metrics, especially diversity. For **Dist-1/2**, LLMs achieve 51.8%[=(2.96-1.95)/1.95] and 92.7%[=(18.29-9.49)/9.49] im-

[2]https://platform.openai.com/docs/models

| Models | Emp. | Coh. | Inf. | Flu. |
|---|---|---|---|---|
| EmpHi | 3.00 | 3.01 | 2.77 | 4.11 |
| KEMP | 2.83 | 2.79 | 2.76 | 4.14 |
| CEM | 3.08 | 3.06 | 2.65 | 4.26 |
| CASE | 3.04 | 3.01 | 2.67 | 4.13 |
| Blenderbot | 3.89 | 3.81 | 3.54 | 4.46 |
| ChatGPT | **4.64** | **4.68** | **4.04** | **4.75** |

Table 2: Results of human ratings about ChatGPT and competitive baselines (the statistical significance (t-test) with p-value < 0.01).

provement, which demonstrates a significant advantage of LLMs in diverse language expression (mainly unigrams and bigrams). In terms of **BERTScore** and **BERT**, LLMs achieve the average improvement of 2.1%[=(2.6+1.6+2.1)/3] and 26.95%[=(18.6+35.3)/2], respectively. This highlights the power of LLMs' in-context learning capability that can be quickly applied to unseen specific tasks. In addition, we observe that the number of exemplars is positively correlated with diversity performance, which suggests the addition of exemplars can influence the linguistic habits of LLMs.

| Models | Dist-1 | Dist-2 | $P_{BERT}$ | $R_{BERT}$ | $F_{BERT}$ | B-2 | B-4 | Acc |
|---|---|---|---|---|---|---|---|---|
| ChatGPT | 2.72 | 17.12 | 0.8679 | 0.8791 | 0.8733 | 6.19 | 1.86 | - |
| + SS ICL | **3.03** | **19.26** | **0.8712** | **0.8804** | **0.8756** | **7.07** | **2.23** | - |
| + Two-stage | 2.51 | 16.90 | 0.8575 | 0.8772 | 0.8671 | 4.99 | 1.37 | **39.18** |
| + Two-stage (emo) | 2.48 | 15.92 | 0.8634 | 0.8772 | 0.8702 | 5.49 | 1.55 | - |
| + Two-stage (situ) | 2.41 | 15.98 | 0.8611 | 0.8774 | 0.8690 | 5.25 | 1.49 | - |
| + Knowledge | 2.73 | 18.29 | 0.8632 | 0.8778 | 0.8703 | 5.31 | 1.47 | - |

Table 3: Results of automatic evaluation on the advanced exploration.

| Comparisons | Emp. | | Coh. | | Inf. | |
|---|---|---|---|---|---|---|
| | Win | Lose | Win | Lose | Win | Lose |
| ChatGPT vs. EmpHi | 95.3% | 0.0% | 91.7% | 0.0% | 95.3% | 0.0% |
| ChatGPT vs. KEMP | 94.0% | 1.3% | 93.3% | 0.3% | 93.7% | 1.0% |
| ChatGPT vs. CEM | 89.3% | 1.3% | 89.0% | 0.7% | 92.7% | 0.3% |
| ChatGPT vs. CASE | 92.7% | 1.3% | 88.7% | 0.7% | 93.3% | 0.7% |
| ChatGPT vs. Blenderbot | 74.7% | 7.7% | 72.3% | 6.0% | 71.0% | 11.0% |
| + SS ICL vs. ChatGPT | 26.3% | 25.3% | 24.3% | 22.0% | 29.7% | 26.3% |
| + Two-stage vs. ChatGPT | 59.3% | 17.3% | 48.0% | 16.0% | 64.7% | 12.7% |
| + Knowledge vs. ChatGPT | 41.7% | 24.3% | 34.7% | 21.0% | 45.0% | 18.3% |

Table 4: Results of human A/B test on aspects (the statistical significance (t-test) with p-value < 0.01).

In the human evaluation, we select ChatGPT (+ 5-shot), which leads in most automatic metrics, as the representative of LLMs. The human ratings and the human A/B test results are listed in Table 2 and Table 4, respectively. We observe ChatGPT also outperforms baselines by a large margin on all aspects, which further demonstrates the outstanding performance of LLMs in generating empathetic, coherent and informative responses. Additionally, we note that the scores of the baselines are lower than in previous studies. This is due to the superior performance of ChatGPT in empathetic dialogue, which relatively raises the standards. It can be corroborated by the fact that in over **70%** of the cases in the A/B test that human annotators prefer responses generated by ChatGPT. For Fluency aspect, there is no significant difference between models, since the responses generated by existing models are already fluent. Therefore, we do not need to compare separately in the A/B test.

## 5.2 Advanced Exploration Results

Experimental results of the advanced exploration on LLMs are shown in Table 3 and the lower part of Table 4. Overall, the responses generated by the three improvement methods we proposed are more favorably received by human annotators in all aspects during the human A/B test, compared to the responses generated by the original Chat-GPT. These results verify the effectiveness of the choice for in-context exemplars, two-stage inter-action generation and enhancement of knowledge related to the context. In the automatic evaluation, the Similar ICL improvement method obtains the best performance, this is attributed to that most automatic metrics tend to favor responses that are closer to the ground truth. However, introducing more thoughts or more information could potentially deviate from the ground truth, even if it's an appropriate response that humans like. Besides, to verify whether the correct inferred emotion or situation has an impact on the generated responses, we conduct two variants experiments of two-stage interactive generation. By separately replacing the model's thinking outputs in the first stage with the truth emotion and situation, results show an enhancement in both BERTScore and BERT metrics. However, this causes a loss in diversity.

## 5.3 Case Study

The generated responses from five competitive baselines and our proposed methods of LLMs are listed in Table 5. It can be observed that most baselines understand the user's feeling, but only pro-

| Emotion | Sad |
|---|---|
| Context | *Speaker*: I think that the girl of my dreams likes somebody else. I feel very sad about it.
*Listener*: ooh, I am so sorry about that. Have you tried to talk to her?
*Speaker*: It's tough because she has been out of the country for a month, so I will likely discuss it when she returns. |
| EmpHi
KEMP
CEM
CASE (SOTA)
Blenderbot | I'm sure she'll be fine.
I am sure you will do great!
Well, you will be fine.
I am sure she will be fine.
That's a good idea. I'm sure you'll be able to figure it out. |
| ChatGPT | That sounds like a good plan. It's important to communicate your feelings and be honest with her. I hope everything works out for you. |
| + SS ICL | That sounds like a good plan. It's important to communicate your feelings and concerns with her. **In the meantime, try to focus on things that make you happy and take your mind off of it.** |
| + Two-stage | I'm sorry to hear that you're **feeling sad about the situation with the girl you like**. It can be tough to navigate these kinds of situations, **especially when the person is out of the country**. Have you thought about how you might approach the conversation when she returns? |
| + Knowledge | It sounds like a tough situation to be in. It's good that **you're planning to talk to her** when she returns. I hope that the conversation goes well and that you can find some clarity and peace of mind. |

Table 5: Generated responses from baselines and LLMs. The bold contents show the effect of improvement methods.

vide simple comforting responses ("*will be fine*"). Blenderbot generates the response with more information while it only supports the user's idea without giving reasons and suggestions. Compared with other baselines, our proposed methods fully understand the user's feeling and generates more empathetic, coherent, and informative responses.

Then we analyze the performance of the improvement methods in this case. The method of semantically similar ICL provides additional suggestions to alleviate the user's sadness emotion ("*focus on things that make you happy*", "*take your mind off*") by learning from relevant instances. The method of two-stage interaction generation reflects inferred user's emotion and situation more specifically in the response. The method of combination with the knowledge base generates the relevant and empathetic response based on the commonsense inference ("*talk to her*") of [*xwant*]. More cases can be found in the Appendix D.

### 5.4 Analysis of LLM Simulating Human Evaluators

LLMs have shown outstanding performance in generating empathetic responses. Naturally, we wonder if it is possible to use LLMs to simulate human evaluators to evaluate the performance of other models. Compared to human evaluators, the latter has lower costs and shorter time consumption. Therefore, we adopt GPT-4 as the evaluator to conduct the A/B test under the same settings. Following Zhong et al. (2022), we use Spearman and

Kendall-Tau correlations to assess the performance of human evaluators and GPT-4. The results are shown in Table 6. We can observe that GPT-4 achieves the best correlation with human evaluators on the aspect of empathy. We observe that GPT-4 has fairly good results in Spearman and Kendall-tau with human evaluators on all aspects (refer to Zhong et al. (2022)), and achieves the best correlation in the aspect of empathy. This indicates the potential of LLMs to simulate human evaluators.

| Model | Aspects | Spearman | Kendall-Tau |
|---|---|---|---|
| GPT-4 | Emp. | 0.485 | 0.467 |
| | Coh. | 0.467 | 0.449 |
| | Inf. | 0.397 | 0.385 |
| | Overall | 0.458 | 0.441 |

Table 6: Spearman and Kendall-Tau correlations of different aspects between human evaluators and GPT-4.

## 6 Conclusion and Future Work

In this work, we empirically study the performance of LLMs on empathetic response generation and propose three improvement methods. Empirical automatic and human evaluation results show that LLMs significantly outperform state-of-the-art models, and verify the effectiveness of our proposed improvements of LLMs.

In the future, our work can contribute to deeper comprehension and the application of LLMs for empathetic dialogue, and provide some insights for similar tasks.

## Acknowledgements

This work is supported by the National Key Research and Development Program (No. 2022YFF0902100) and National Natural Science Foundation of China (No. 62076081 and No. 61936010).

## Limitations

The main limitation of our work is the shortage of standard datasets in the task of empathetic response generation. Although there are efforts (Welivita et al., 2021) to construct relevant datasets, their quality and popularity are far inferior to EM-PATHETICDIALOGUES. Another limitation is that empathy is a complex concept, and different personalities, backgrounds, and cultures may have different ways of expressing empathy. However, our work do not consider the above factors, and we will explore them in the future work.

## Ethics Statement

The dataset we adopt in this paper is a publicly available corpus. The EmpatheticDialogues dataset is annotated by Amazon Mechanical Turk, and the dataset provider filters all personal information and unethical language. We belieive that this work complies with the ethical policy of EMNLP.

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

## A Prompt Template

Table 7 shows the example of the prompt template.

## B Comparable Models Details

The following are the models we compared in the experiments. We use the official codes and follow the implementations.

(1) MoEL (Lin et al., 2019): A Transformer-based model that employs multiple emotion specific decoders to generate empathetic responses [3].

(2) MIME (Majumder et al., 2020): A Transformer-based model that explicitly targets empathetic dialogue by leveraging polarity-based emotion clusters and emotion mimicry [4].

(3) EmpDG (Li et al., 2020): An interactive adversarial model which exploits the user feedback, the coarse-grained dialogue-level, and fine-grained token-level emotions [5].

(4) EC (Gao et al., 2021): Employing the identification of emotion causes of the context and gated mechanism to enhance the generation of empathetic responses. There are soft and hard gated mechanisms [6].

(5) EmpHi (Chen et al., 2022): Employing a discrete latent variable to understand the distribution of potential empathetic intentions, and integrating

---

[3] https://github.com/HLTCHKUST/MoEL
[4] https://github.com/declare-lab/MIME
[5] https://github.com/qtli/EmpDG
[6] https://github.com/A-Rain/EmpDialogue_RecEC

| Prompt Template | Contents |
|---|---|
| Task Definition | *This is an empathetic dialogue task: The first worker (Speaker) is given an emotion label and writes his own description of a situation when he has felt that way. Then, Speaker tells his story in a conversation with a second worker (Listener). The emotion label and situation of Speaker are invisible to Listener. Listener should recognize and acknowledge others' feelings in a conversation as much as possible.* |
| Guideline Instruction | *Now you play the role of Listener, please give the corresponding response according to the existing context. You only need to provide the next round of response of Listener.* |
| Exemplars | *The following is the existing dialogue context:* 
 *Instance 1:* 
 (the complete dialogue from the training set...) |
| Dialogue Context | *Speaker: $U_1$* 
 *Listener: $U_2$* 
 *......* 
 *Speaker: $U_{n-1}$* |
| Others | The additional contents for improvement methods. |

Table 7: The example of the prompt template.

implicit and explicit intent representations to produce empathetic responses [7].

(6) KEMP (Li et al., 2022): A model leverages external knowledge, including commonsense and emotional lexical knowledge, to explicitly understand and express emotions in empathetic dialogue generation [8].

(7) CEM (Sabour et al., 2022): A new approach leverages commonsense knowledge, combining affective and cognitive aspects, to further enhance empathetic expressions in generated responses [9].

(8) CASE (Zhou et al., 2022): Align users' cognition and affection at coarse-grained and fine-grained levels through the commonsense cognitive graph and emotional concept graph [10].

(9) BlenderBot (Roller et al., 2021): An open-domain chatbot with a number of skills. We adopt BlenderBot 90M, which was fine-tuned on the EM-PATHETICDIALOGUES train dataset [11].

## C   Human Evaluation Details

For each case, we provide the annotators with the dialogue context and responses from compared models. The responses of compared models in each case are randomly disrupted, so the evaluators are not affected by the order of the models.

## D   Additional Case Study

Table 8 shows additional case study of generated responses from SOTA baselines and our proposed methods.

---

[7] https://github.com/mattc95/EmpHi
[8] https://github.com/qtli/KEMP
[9] https://github.com/Sahandfer/CEM
[10] https://github.com/jfzhouyoo/CASE
[11] https://parl.ai/projects/recipes

| Emotion | Apprehensive |
|---|---|
| Context | *Speaker*: I was so scared of going onto the train!
*Listener*: Was it your first time riding on one? Is there something in particular that makes you afraid?
*Speaker*: No but I hate going in objects that travel fast, I really did not want to go on it. |
| EmpHi | What a lot of fun. What kind of car do you have? |
| KEMP | I am sure you will not have any time to get out of it. |
| CEM | I am glad you are okay. |
| CASE (SOTA) | I wouldn't be too scared of them too. |
| Blenderbot | I'm sorry to hear that. It's always good to be prepared for things like that. |
| ChatGPT | I understand how you feel. It can be scary to be in a situation that makes you uncomfortable.
Did anything help you feel better while you were on the train? |
| + SS ICL | I can understand why you would feel that way. **It can be scary to be in a fast-moving object, especially if you're not used to it.** Did anything happen on the train that made you feel better or worse? |
| + Two-stage | I can understand why you would **feel scared in that situation**. **It can be really overwhelming to be in a fast-moving object like a train**. Did anything in particular happen that made you feel more scared than usual? |
| + Knowledge | I understand how you feel. It's completely normal to **feel scared** when you're doing something that makes you uncomfortable. Did anything specific happen on the train that made you **feel scared**, or was it just the speed of the train that made you feel uneasy? |
| Emotion | Sentimental |
| Context | *Speaker*: I found an old photo album at my grandma's house that had pictures of me when I was a kid.
*Listener*: That's cool, did it bring back some good memories?
*Speaker*: Yeah, it did. I really miss the 90s. Everything seemed a lot simpler in those days. |
| EmpHi | That's cool. I bet it was fun to see. |
| KEMP | I am sure you will do great! |
| CEM | I bet it was a great time to go back and see. |
| CASE (SOTA) | It is always good to see old memories of it. |
| Blenderbot | I'm sorry to hear that. it's always nice to see the good old days. |
| ChatGPT | I know what you mean. It's nice to look back on old memories and remember the good times. |
| + SS ICL | I know what you mean. It's nice to look back on **those simpler times and remember the good moments**. |
| + Two-stage | That sounds like a really special moment. **It's nice to be able to look back on happy memories and feel that sense of nostalgia**. What was your favorite memory from that time? |
| + Knowledge | It's nice that you were able to find something that brought back happy memories and made you **feel nostalgic**. Sometimes it's good to take a trip down memory lane and **remember the good old days**. |

Table 8: Generated responses from baselines and LLMs. The bold contents show the effect of improvement methods.