# OpenReview forum: "Harnessing the Power of Large Language Models for Empathetic Response Generation: Empirical Investigations and Improvements"
_EMNLP/2023/Conference — EMNLP 2023 Findings_

### Official Review · Reviewer_UTmi · 2023-08-05

**Soundness:** 4

**Excitement:**

3: Ambivalent: It has merits (e.g., it reports state-of-the-art results, the idea is nice), but there are key weaknesses (e.g., it describes incremental work), and it can significantly benefit from another round of revision. However, I won't object to accepting it if my co-reviewers champion it.

**Paper Topic And Main Contributions:**

Topic: This paper studies empathetic dialogue generation leveraging large language models, in particular, ChatGPT.

Research problem: The backbone models adopted by previous studies on empathetic dialogue generation are generally small in scale limiting their empathetic and language understanding capability. The recently-emerged ChatGPT has demonstrated extraordinary performance in a wide variety of tasks. Yet, its capability in the task of empathetic response generation remains underexplored.

Contributions: The authors propose three different methods to augment ChatGPT so as to boost its empathetic generation capability compared to zero-shot inference: (1) retrieve semantically similar exemplars for in-context learning; (2) two-stage interaction, i.e, a chain-of-thought style of prompt whereby the model is first instructed to infer the emotion and situational contexts, next, based on the inferred emotion and situational contexts, the model is further instructed to generate the empathetic response; (3) augment the dialogue context with external commonsense knowledge-base, similar to what has been proposed in [1].

[1] Sabour, Sahand, Chujie Zheng, and Minlie Huang. "CEM: Commonsense-aware empathetic response generation." Proceedings of the AAAI Conference on Artificial Intelligence. Vol. 36. No. 10. 2022.

**Questions For The Authors:**

(1) Line 317, do you concatenate commonsense inferences with respect to all the five relationship types with the dialogue context? If so, how do you know that all the relationship types are relevant to the dialogue context?

(2) I don’t see the algorithm you mentioned in Line 318 - 312, “We also design an algorithm to construct the suitable prompt ….”

(3) Your analyses are only limited to ChatGPT. How about other LLMs, will the ideas be generally applicable to other LLMs?

**Reasons To Accept:**

(1) The experiments are thorough and demonstrate the effectiveness of the authors’ proposed ideas.

(2) The clarity of the paper is good.

**Reasons To Reject:**

The contribution is incremental. The proposed ideas to augment ChatGPT have been extensively studied in prior works for other tasks. For example, using semantically similar exemplars for in-context learning [2,3], CoT prompting [4,5], and using external commonsense knowledge [1]. The authors should highlight the motivation of applying such ideas in the context of empathetic dialogue generation, the unique challenges of empathetic dialogue generation, and why applying such ideas can help address the challenges.

[1] Sabour, Sahand, Chujie Zheng, and Minlie Huang. "Cem: Commonsense-aware empathetic response generation." Proceedings of the AAAI Conference on Artificial Intelligence. Vol. 36. No. 10. 2022.

[2] Rubin, Ohad, Jonathan Herzig, and Jonathan Berant. "Learning To Retrieve Prompts for In-Context Learning." Proceedings of the 2022 Conference of the North American Chapter of the Association for Computational Linguistics: Human Language Technologies. 2022.

[3] Liu, Jiachang, et al. "What Makes Good In-Context Examples for GPT-3?." Proceedings of Deep Learning Inside Out (DeeLIO 2022): The 3rd Workshop on Knowledge Extraction and Integration for Deep Learning Architectures. 2022.

[4] Kojima, Takeshi, et al. "Large language models are zero-shot reasoners." Advances in neural information processing systems 35 (2022): 22199-22213.

[5] Wei, Jason, et al. "Chain-of-thought prompting elicits reasoning in large language models." Advances in Neural Information Processing Systems 35 (2022): 24824-24837.

**Reproducibility:**

4: Could mostly reproduce the results, but there may be some variation because of sample variance or minor variations in their interpretation of the protocol or method.

**Reviewer Confidence:**

4: Quite sure. I tried to check the important points carefully. It's unlikely, though conceivable, that I missed something that should affect my ratings.

---

> ### Author Rebuttal · Authors · 2023-08-28
>
> We sincerely thank you for your time and efforts!  We will explain your concerns point by point.
>
> Question 1:
>
> Yes, the commonsense inferences of the five relationships are concatenated with the dialogue context. The **commonsense inferences are obtained dynamically based on different dialogue contexts**, or 'none' if not present. The obtained commonsense knowledge **is related to the context, but the degree of relevance between different types of knowledge and dialogue contexts varies**. This tests the reasoning ability of LLMs, and how to use more appropriate knowledge to generate responses. If LLMs are not good at selecting relevant knowledge, we can also filter the knowledge with other small models that are good at this ability, and then provide it to LLMs.
>
> Question 2:
>
> **Our algorithm will obtain corresponding commonsense inferences based on different dialogue contexts and dynamically concatenate them into the prompt**. Therefore, each instance in the test set will have a suitable prompt. The composition of the prompt remains unchanged, but its content will change accordingly. We will explain it more clearly in the revised version. Our code will be open source.
>
> Question 3:
>
> Our main experimental LLMs **include ChatGPT (gpt-3.5-turbo), GPT-3.5 (text-davinci-003), and GPT-3 (davinci)**. During the human evaluation, considering the cost of manual labor, we chose ChatGPT as the representative of LLMs, which is of epochal significance and has excellent automatic evaluation results. **The ideas can be applicable to other LLMs, since ICL, CoT, and instruction following are common capabilities that LLMs of a certain scale possess**. Of course, the strength of these capabilities will be different, and it is worthy of further experimental exploration in the future.
>
> About the motivations for the improvement methods, in each part of Section 3.3, we **have a brief description, specifically in Lines 220-225, Lines 256-265, and Lines 287-297**. However, as you pointed out, we may need to integrate more closely with relevant theoretical foundations.

---

### Official Review · Reviewer_emhy · 2023-08-08

**Typos Grammar Style And Presentation Improvements:** 1. Tables 5 and 8 should include the …
**Soundness:** 4

**Excitement:**

3: Ambivalent: It has merits (e.g., it reports state-of-the-art results, the idea is nice), but there are key weaknesses (e.g., it describes incremental work), and it can significantly benefit from another round of revision. However, I won't object to accepting it if my co-reviewers champion it.

**Missing References:**

1. This paper uses a technique related to in-context learning with exemplars: Majumder, Navonil, et al. "Exemplars-guided empathetic response generation controlled by the elements of human communication." IEEE Access 10 (2022): 77176-77190.

2. A concurrent work on empathetic response generation with LLMs: Zhao, Weixiang, et al. "Is ChatGPT Equipped with Emotional Dialogue Capabilities?." arXiv preprint arXiv:2304.09582 (2023).

**Paper Topic And Main Contributions:**

The paper presents a set of methodologies for empathetic response generation with large language models (LLMs). OpenAI’s GPT family models, which often have excellent conversational skills, are used as the LLM. The paper benchmarks the following strategies for improved empathetic response generation: i) semantically similar in-context learning, ii) two-stage interactive generation, and iii) improvement with commonsense knowledge bases. The proposed methods are evaluated on the widely used Empathetic Dialogues dataset.

  The paper is easy to follow and tackles an important problem. The evaluation could be more thorough and will add more value to the paper. Some relevant references are missing. The result is intuitive and probably expected, as the LLMs used in the paper have shown excellent results in many related tasks. However, it’s always good to have a thorough study with a comprehensive evaluation. In that sense, I think the paper has thoughtful experiments, good evaluation, and solid contributions overall.

**Questions For The Authors:**

1. Are the evaluation metrics biased toward the length of the generated output? It seems like human evaluators prefer generated outputs that are lengthier. Could you try limiting the output length from LLMs to the average target reference length in the dataset and recheck the numbers?

2. The ChatGPT + SS ICL shows the best automatic eval metrics in Table 3 but is less preferred by human evaluators compared to the other strategies in Table 4 (two-stage knowledge). It would be great to know the author's thoughts about this.

 3. It seems like there are many ties in the A/B testing for i) ChatGPT + SS ICL vs. ChatGPT and ii) ChatGPT + Knowledge vs. ChatGPT. Can you discuss why this is the case?

4. For in-context learning, did you select a consistent set of exemplars for all the test set (based on average similarity from equation 4), or did you select different exemplars for different instances of the test set?


**Reasons To Accept:**

1. Clear discussion of problem statement and proposal of three fundamental techniques for improved empathetic response generation.
2. Improved results over several baseline methods in both automatic and human eval metrics.
3. The examples shown in paper indicate that the proposed method is powerful and could be explored further for the problem of empathetic response generation.

**Reasons To Reject:**

1. A weakness is that the evaluation could have been more through. Although expected, it is unclear which components bring the improvement in performance. Are the in-context example responses very similar to the target test references? What happens when you try to use all three of the proposed techniques together (ChatGPT + SS ICL + Two-stage + Knowledge)?
2. Some important details about the proposed components could have been added to the paper. For example, how does the generated knowledge snippets from COMET in section 3.3.3 look like?

**Reproducibility:**

4: Could mostly reproduce the results, but there may be some variation because of sample variance or minor variations in their interpretation of the protocol or method.

**Reviewer Confidence:**

3: Pretty sure, but there's a chance I missed something. Although I have a good feel for this area in general, I did not carefully check the paper's details, e.g., the math, experimental design, or novelty.

---

> ### Author Rebuttal · Authors · 2023-08-28
>
> We sincerely thank you for your time and efforts!  We will explain your concerns point by point.
>
> Question 1:
>
> In fact, before conducting the experiments, we had thought that the output length of LLMs may have an impact. Initially, we considered calculating the average length of target responses in the training set to limit the output length of LLMs. We **come up with two methods: one is to set the parameter "max_tokens" when calling the API, and the other is to add a description of the length limit in the prompt. However, both of the above methods have problems**. Most directly, **the number of tokens is not equal to the number of words**, which depends on the tokenization algorithm adopted by the LLMs. The method of setting the 'max_tokens' parameter will cause LLMs to **directly truncate** the output upon reaching the maximum number of tokens, resulting in semantically incomplete outputs. The method of limiting the output length through the prompt is influenced by **the ability of LLMs to follow instructions**, and the actual response length obtained may also **deviate** from the required one. Furthermore, if the length of the output directly affects human preferences, then designing small language models (SLMs) can also consider producing longer responses. In summary, we ultimately chose the method of stopping the output when LLMs generate the termination token.
>
> Question 2:
>
> Most automatic metrics currently used are based on **the proximity to the ground truth**. However, the response that satisfies users in open-domain dialogue is not unique, which is also a widespread problem in open-domain dialogue. Human evaluation is closer to real users but costly and time-consuming. Given the outstanding performance of LLMs, this paper also explores the possibility of using GPT-4 to simulate human evaluators. The attempt is preliminary but has shown potential.
>
> Question 3:
>
> In the A/B test, there are indeed many ties between ChatGPT and improvement methods. We speculate that this is because **when the results have reached the satisfaction standard of the evaluators, they tend not to make excessive distinctions**. How to set up a more fine-grained and discerning human evaluation method is a valuable and meaningful question.
>
> Question 4:
>
> We compute the semantic similarity between each instance (**excluding ground truth responses**) in the test set and each instance in the training set. For each instance in the test set, related exemplars are **automatically selected** based on the similarity values. That means it's **the latter**.
>
> About the missing references, thank you for pointing them out. We will add them to the revised version in time.
>
> About the presentation improvements, thank you for your advice. We will add the annotated target responses from the original dataset to the tables showing the case study.
>
> **About the generated knowledge snippets from COMET**, taking Table 5 as an example, the knowledge inferences generated by this instance are:
>
> x_intent:['to talk to her', 'to make a decision', 'to talk about it', 'to talk to someone', 'to discuss the issue']
>
> x_need:['to talk to her', 'to get to know her', 'to talk to her .', 'to have a conversation', 'to talk to someone']
>
> x_want:['to talk to her', 'to talk about it', 'to make a decision', 'to have a conversation', 'to talk to someone']
>
> x_effect:['gets stressed out', 'get a job', 'none', 'gets stressed', 'gets nervous']
>
> x_react:['relieved', 'happy', 'sad', 'nervous', 'anxious']

---

### Official Review · Reviewer_h8tQ · 2023-08-12

**Soundness:** 4

**Excitement:**

3: Ambivalent: It has merits (e.g., it reports state-of-the-art results, the idea is nice), but there are key weaknesses (e.g., it describes incremental work), and it can significantly benefit from another round of revision. However, I won't object to accepting it if my co-reviewers champion it.

**Paper Topic And Main Contributions:**

This paper investigates using more recent large pre-trained language models for empathetic response generation.  They show that basic chatgpt adapted for this task improves on the current SOTA and that this can be further improved with enhancement via: (1) similar in-context learning, (2) two-stage reasoning about the conversation before generating and (3) by adding external knowledge graph information.  They show improvements using both automatic and human metrics.

**Questions For The Authors:**

Is the \oplus in equation 7 meant to signify concatenation in this case?

**Reasons To Accept:**

- Show improvement over existing sota on this benchmark
- Introduce three methods of improving chatgpt for this task which seem to be mostly successful in improving performance
- Comprehensive automatic and human evaluations.  Experiments and metrics seem reasonable
- Generated output in Table 5 looks very impressive

**Reasons To Reject:**

- Some of the results tables claim that statistical significance testing has taken place, but it’s unclear which things are being compared in the statistical tests.   This seems important to clarify since some of the differences are a bit smaller.
- This paper is empirically significant in establishing a new SOTA on this benchmark, but the technical novelty of the approach is more limited.  It’s not proposing a new modelling approach, and the ideas for improving over chatgpt (e.g. two stage reasoning with emotion prediction, using a knowledge base, etc) are similar to prior approaches that have been successful in improving smaller models on this dataset.

**Reproducibility:**

5: Could easily reproduce the results.

**Reviewer Confidence:**

4: Quite sure. I tried to check the important points carefully. It's unlikely, though conceivable, that I missed something that should affect my ratings.

---

> ### Author Rebuttal · Authors · 2023-08-28
>
> We sincerely thank you for your time and efforts!  We will explain your concerns point by point.
>
> **About the statistical significance testing**, we tested all the human evaluation metrics, and the p-value is less than 0.01. We employed **the ''ttest_ind'' function of the ''scipy.stats'' library**, and our evaluation code and results can be published. Regarding the automatic evaluation results, we can observe **the internal comparison of the SOTA series (the upper part of Table 1)**, and our improvement is extremely significant.
>
> **About the technical novelty**, our improvement methods for LLMs mainly adopt **gradient-free techniques**, which are based on the paradigm shift brought by the emerging capabilities (ICL, CoT, etc.) of LLMs. This has been rarely studied in past empathetic dialogue work. We have **designed a prompt framework** for LLMs to apply in the field of empathetic dialogue. Among them, prompts can be **automatically and adaptively** changed according to different methods.
>
> **About the question**, yes, the \oplus in equation 7 means the concatenation of the five corresponding commonsense inferences. We will add this explanation to the revised version.

---

### Official Review · Reviewer_LJBU · 2023-08-12

**Typos Grammar Style And Presentation Improvements:** 1. Would be helpful to add * indicati…
**Soundness:** 4

**Excitement:**

4: Strong: This paper deepens the understanding of some phenomenon or lowers the barriers to an existing research direction.

**Missing References:**

1. Ashish Sharma, Adam S Miner, David C Atkins, and Tim Althoff. 2020. A computational approach to understanding
empathy expressed in text-based mental health support. arXiv preprint arXiv:2009.08441.

**Paper Topic And Main Contributions:**

In this work, the authors conduct a comprehensive analysis of empathetic response generation and comparison of 3 proposed methods to improve empathetic dialogue: selecting in-context examples using semantic similarity, using chain of thought reasoning in speculating the user's emotions, and augmenting knowledge using knowledge graphs. The authors conduct both automatic and human evaluation across a variety of empathetic dialogue metrics and additionally assess the viability of using LLMs such as GPT-4 to simulate human empathy evaluations.

**Questions For The Authors:**

A. Did you ask human evaluators to provide rationales for why certain dialogues were more empathetic than others? This would be interesting supplementary analysis to further interpret the contribution of the 3 proposed improvements to empathetic response generation.

B. How did combining the 3 improvement methods influence overall performance?

**Reasons To Accept:**

Overall, the paper is well-organized, well-written, and the evaluation is quite comprehensive. Empathetic chat is a task that has garnered growing interest in recent years, and, as such, this work is valuable in serving as an evaluation benchmark on empathic quality of LLM-generated text. In addition, the three proposed improvements to prompt-based empathetic dialogue generation are well-validated, and both the human and automatic evaluation are well-designed and straightforward. The paper provides clear qualitative examples of the effects the author's proposed improvements have on empathetic dialogue.

**Reasons To Reject:**

1. The main limitation of this work, as the authors mention, is the focus on evaluation with only the EmpatheticDialogues dataset. Conducting similar evaluations across more diverse datasets would be needed for more generalizable results.
2. The human evaluation has few annotators, which can bias the evaluation towards a narrow definition of empathy.
3. Within the human evaluation, the provided instructions on ranking empathy are not clear, and it appears that sub-dimensions of empathy are not included in the evaluation, which could improve interpretability of the results.
4. Discussion of the analysis on using LLMs to simulate human evaluators is cursory, and would warrant more analysis beyond the scope of the contributions of this work. While this evaluation is timely, correlation alone cannot demonstrate that LLMs effectively simulate empathy reasoning, and the results of this analysis do not clearly bolster the contributions of this work.

**Reproducibility:**

4: Could mostly reproduce the results, but there may be some variation because of sample variance or minor variations in their interpretation of the protocol or method.

**Reviewer Confidence:**

5: Positive that my evaluation is correct. I read the paper very carefully and I am very familiar with related work.

---

> ### Author Rebuttal · Authors · 2023-08-28
>
> We sincerely thank you for your time and efforts!  We will explain your concerns point by point.
>
> Question A:
>
> Great idea! This will **aid in the research on interpretability and the exploration of theoretical validation**. It's a pity we didn't take this into account during human evaluation. As a supplement, in our experiment of simulating human evaluators with GPT-4, we found that **GPT-4 can provide its rationales**. We believe this will help in verifying the rationality of its judgments in the future.
>
> Question B:
>
> We have yet to consider the possibility of combining the three methods. Thank you for your inspiration. Intuitively, they cannot be directly combined, and there may be situations of mutual influence. **How to combine the three methods in an appropriate way is a valuable exploration work** that may require more theoretical research, and we will explore it in the future.
>
> About the missing reference, thank you for pointing it out. We will add it to the revised version in time. Thank you for your suggestion on presentation improvements.

---

### Official Review · Reviewer_5Xfj · 2023-08-14

**Soundness:** 3

**Excitement:**

3: Ambivalent: It has merits (e.g., it reports state-of-the-art results, the idea is nice), but there are key weaknesses (e.g., it describes incremental work), and it can significantly benefit from another round of revision. However, I won't object to accepting it if my co-reviewers champion it.

**Paper Topic And Main Contributions:**

The paper studies how Large Language Models (LLMs) like ChatGPT perform on the empathetic dialogue task. The authors use automatic metrics and human evaluations to show that ChatGPT outperforms current sota models on this benchmark. They also suggest several strategies to boost ChatGPT's base performance further.

**Questions For The Authors:**

A. Line 57: Can you clarify the limitations of the approach you mentioned and distinguish what your paper offers as an alternative?

B. Line 65: The abbreviation "ICL" is introduced without a prior explanation. Can you rearrange its introduction for clarity?

C. Line 202: How did you decide on the chosen prompt? Were other prompt variations considered or tested?

D. Line 227: By selecting exemplars that closely match the test set, aren't you artificially enhancing the performance metrics? Typically, similar instances are reserved for one split to gauge genuine performance.

E. Line 235: What led to the choice of the particular embedding mentioned?

F. Line 276: For the method mentioned, were alternative formulations or variations tested in an ablation study?

G. Line 356: What motivated the decision to use numerous metrics? Given that many of these metrics often correlate, and some may not be reliable for longer texts, a justification would be helpful.

H. Table 1: Given the extensive results, would the differences remain statistically significant if corrections (like the Bonferroni correction) were applied for multiple comparisons?

I. Line 453: For the human evaluation, were assessors shown both versions of the answers? If so, in what sequence? More details about the human evaluation procedure would be beneficial.

K. Line 553: This line seems to repeat information from the previous sentence. Can you clarify?

**Reasons To Accept:**

* The paper offers a detailed examination of LLMs' performance in the domain of empathetic dialogue, providing valuable insights into their capabilities.
* The methods and templates introduced by the authors can serve as valuable tools for researchers and practitioners in the NLP community.
* The motivation behind the study is well-articulated, emphasizing the real-world relevance of the task. Additionally, the related work section effectively lays down foundational knowledge

**Reasons To Reject:**

*  Some of the methodologies in the paper, like the "tricky" choice of exemplars for ICL that matches most closely with the test set, raise concerns. This potentially makes the improvements and all subsequent results unreliable.
* The paper lacks crucial details about the human evaluation process. Information regarding the hiring process, demographic details of the evaluators, measures taken to ensure unbiased evaluations, and procedures to validate the accuracy of annotations is missing.
* The paper does not specify if the proposed prompt template was compared against other prompt types or variations.

**Reproducibility:**

4: Could mostly reproduce the results, but there may be some variation because of sample variance or minor variations in their interpretation of the protocol or method.

**Reviewer Confidence:**

4: Quite sure. I tried to check the important points carefully. It's unlikely, though conceivable, that I missed something that should affect my ratings.

---

> ### Author Rebuttal · Authors · 2023-08-28
>
> We sincerely thank you for your time and efforts!  We will explain your concerns point by point.
>
> Question A:
>
> A clearer description of this question is in **Lines 147-160**. Compared with [1], our work mainly adopts some gradient-free techniques, which also represent the exploration of the paradigm shift. Compared with [2] only conducting experiments on GPT-3, we explored GPT-3, GPT-3.5, and ChatGPT. Furthermore, in addition to the in-context learning capacity of LLMs, we also explored the capacity of chain-of-thought and combining external knowledge. Empirical Investigations are more **comprehensive** and the perspective is **broader**.
>
> Question B:
>
> ''ICL'' is the abbreviation of "in-context learning". Thank you for pointing out this issue. We will **add it after "in-context learning" on Line 51** of the introduction in the revised version.
>
> Question C:
>
> As we know, LLMs are sensitive to prompts. Therefore, when we consider the construction of prompts, we adhere to **the principle of clearly expressing requirements, providing sufficient context, and enabling the language model to accurately understand our intentions**. In practice, we **iteratively optimize** the prompt on a small number of samples, and finally select the prompt template that enables LLMs to follow instructions accurately on samples. After our code is open-sourced, we will **display all the prompts employed** (The prompt template remains the same, and there will be differences in content when different methods are involved).
>
> Question D:
>
> Firstly, the data we use comprises the training set and the context of the test set (**excluding ground truth responses**), which are visible to all baseline models. This means it is **fair**. Secondly, our semantic similarity matching is executed automatically by the code, which means it is **efficient**. Additionally, our method theory has also been **validated through previous work** [2, 3].
>
> Question E:
>
> Because the "all-mpnet-base-v2" model provides **the best quality** in the extensive experiments, which is described in **SentenceTransformers** documentation (The initial work is [4]). Given that links are not allowed in the rebuttal, we will **add the related footnote** in the revised version.
>
> Question F:
>
> Yes, we conducted two variants experiments of two-stage interactive generation to explore the impact of different key factors (emotions and situations) on the results, **described in Lines 481-489**, and the results are **shown in Table 3**.
>
> Question G:
>
> For the choice of evaluation metrics, we primarily **follow previous related studies** [5-8] and choose **the union** as much as possible. The reason is to evaluate the performance of LLMs in the empathetic response task more comprehensively and fairly. It's worth noting that we did not adopt the commonly used metric PPL, due to differences in vocabulary across models. Moreover, LLMs tend to significantly outperform ordinary small models on this metric.
>
> Question H:
>
> Our work has significantly improved compared to previous research, which can be **verified through internal comparisons in the upper half of Table 1**. The work in the state-of-the-art baselines section is basically in chronological order, among which, CASE is the latest SOTA in the empathetic dialogue domain, accepted by ACL 2023. Compared to the improvement of the latest SOTA over previous work, our improvement is significant. Regarding the corrections you mentioned, we will consider using them after careful study.
>
> Question I:
>
> The content in this line is not clear enough, so let's provide more details. Firstly, it should be clarified that both the research on the models we compared and the dataset used in this paper are the benchmark (EmpatheticDialogues), and **the test set is consistent. When we run baselines, we employ the official codes and recommended parameters**. The generated results can basically match the values provided in the original papers in terms of automatic metrics. However, there are differences between the manual metrics and those provided in the original papers. Specifically, in the research on baselines, the scores of manual metrics are generally above 4, and the win rate in the A/B test doesn't exceed 60% comprehensively. We speculate that this is because the responses generated by LLMs have **increased the upper limit**, and in comparison, the results of baselines have been passively decreased.
>
> Question K:
>
> Because the contribution of these works is the construction of a standard dataset for empathetic dialogue. However, when we conducted experiments on these datasets, we found that the quality is indeed inferior to the benchmark dataset (EmpatheticDialogues). This is understandable because the method is semi-automatic and provides a larger scale. Nevertheless, constructing large-scale high-quality datasets still requires more work.
>
> **About the human evaluation process**, we have described human evaluation details in **Appendix C**, and we will provide further details. The evaluators are above a bachelor's degree, two men and one woman. **Before the evaluation**, we have specific evaluation instructions, and we will select 10% of the parts to verify whether they understand the requirements. **During the evaluation process**, the response of each model will be shuffled in each round to remove the influence of order on the evaluation. **After the evaluation**, we employ the "ttest_ind" function of the "scipy.stats" library and  Fleiss’ Kappa to verify significance and consistency, respectively.
>
> For concerns about the choice of exemplars for ICL, please refer to Question D.
>
> For concerns about the proposed prompt template, please refer to Question C.
>
> [1] Roller, Stephen, et al. "Recipes for Building an Open-Domain Chatbot." Proceedings of the 16th Conference of the European Chapter of the Association for Computational Linguistics: Main Volume. 2021.
>
> [2] Lee, Young-Jun, Chae-Gyun Lim, and Ho-Jin Choi. "Does GPT-3 generate empathetic dialogues? A novel in-context example selection method and automatic evaluation metric for empathetic dialogue generation." Proceedings of the 29th International Conference on Computational Linguistics. 2022.
>
> [3] Liu, Jiachang, et al. "What Makes Good In-Context Examples for GPT-3?." Proceedings of Deep Learning Inside Out (DeeLIO 2022): The 3rd Workshop on Knowledge Extraction and Integration for Deep Learning Architectures. 2022.
>
> [4] Reimers, Nils, and Iryna Gurevych. "Sentence-BERT: Sentence Embeddings using Siamese BERT-Networks." Proceedings of the 2019 Conference on Empirical Methods in Natural Language Processing and the 9th International Joint Conference on Natural Language Processing (EMNLP-IJCNLP). 2019.
>
> [5] Majumder, Navonil, et al. "MIME: MIMicking Emotions for Empathetic Response Generation." Proceedings of the 2020 Conference on Empirical Methods in Natural Language Processing (EMNLP). 2020.
>
> [6] Gao, Jun, et al. "Improving empathetic response generation by recognizing emotion cause in conversations." Findings of the association for computational linguistics: EMNLP 2021. 2021.
>
> [7] Sabour, Sahand, Chujie Zheng, and Minlie Huang. "Cem: Commonsense-aware empathetic response generation." Proceedings of the AAAI Conference on Artificial Intelligence. Vol. 36. No. 10. 2022.
>
> [8] Zhou, Jinfeng, et al. "CASE: Aligning Coarse-to-Fine Cognition and Affection for Empathetic Response Generation." arXiv preprint arXiv:2208.08845 (2022).

---

### Meta-Review · Area_Chair_vHWQ · 2023-10-06

**Recommendation:** 5

**Metareview:**

The research makes a good contribution for empathetic quality in LLMs text. The paper presents a comprehensive automatic and human evaluations. The qualitative examples provided further illustrate the impact of the proposed improvements on empathetic dialogue.

---

### Decision · Program_Chairs · 2023-10-07

**Decision:**

Accept-Findings

**Comment:**

The research makes a good contribution for empathetic quality in LLMs text. The paper presents a comprehensive automatic and human evaluations. The qualitative examples provided further illustrate the impact of the proposed improvements on empathetic dialogue.